# Foundations without Fundamentals: Zero-Shot Blind Spots in Time Series FMs

**Nafiseh Ghoroghchian** [1]  **Haipeng Zhang** [1]  **Shuyi Han** [1]  **Alex Labach** [1]  **George Stein** [1]

## Abstract

Despite the success of Time Series Foundation Models (TSFMs) on broad benchmarks, their ability to internalize basic temporal logic, especially in settings supported by exogenous covariates, remains under-examined. We introduce SimpleTimeBench[1], a diagnostic univariate and multivariate "unit test" suite for primitives such as monotonic trends, periodic signals and leading indicator covariates, scenarios where near-perfect forecasts should be trivial. Surprisingly, prominent multivariate TSFMs (Chronos-2, Moirai and Toto) frequently produce suboptimal zero-shot forecasts for these inputs. While fine-tuning Chronos-2 improves its behaviour on specific tasks, we show that this adaptation degrades performance on other fundamental patterns rather than enhancing its generalizable foundational capabilities. This reveals a gap between pre-training scale and basic temporal reasoning, suggesting that current TSFMs could potentially lack the inductive biases needed to capture simple predictable functions. We further demonstrate that these failures are not merely synthetic curiosities: they persist in real-world sensor forecasting, where TSFMs consistently underutilize leading indicators available in observed covariates. This inability to capture simple relationships limits the practical utility and reliability of current multivariate models.

## 1. Introduction

Time series foundation models (TSFMs) continue to gain traction and achieve success on public benchmarks such as GIFT-Eval (Aksu et al., 2024) and fev-bench (Shchur et al., 2025), aiming to cover a diverse range of real-world forecasting settings in both univariate and multivariate scenarios. While early foundational efforts mainly focused on univariate approaches, architectures built to utilize multivariate dependencies such as Chronos-2 (Ansari et al., 2025), Toto (Cohen et al., 2024) and Moirai (Woo et al., 2024) promise broader applicability in complex real-world scenarios where system dynamics are driven by exogenous factors. These benchmarks are typically assessed using aggregate metrics (e.g., CRPS or MASE) averaged over thousands of series. While critical for pushing the performance frontier, these aggregate measures and benchmarks do not clearly identify fundamental weaknesses.

We posit that, just as LLMs exhibit a *jagged frontier* by excelling at complex tasks while failing at simple ones (Nezhurina et al., 2025; Dziri et al., 2023), TSFMs may possess similar systematic blind spots. To probe this, we evaluate TSFMs through the lens of "time series unit tests" - simplistic tasks where the data-generating process is transparent and expected performance is near-perfect for classical statistical models or even human interpolation. To truly serve as foundational backbones, TSFMs must be able to extrapolate trends, continue periodic signals, and exploit strong Granger-causal signals where a covariate is a leading indicator, i.e., clearly indicates future target values.

In this work, we introduce SimpleTimeBench, a combined univariate, multivariate, and covariate-supported diagnostic suite designed to isolate these fundamental capabilities. By stripping away the noise of complex datasets, we evaluate simple functional capabilities of TSFMs, such as monotonic extrapolation and identity mapping, where the correct forecast is mathematically trivial. Our evaluation reveals zero-shot blind spots: despite their sophisticated architectures, Chronos-2, Moirai and Toto struggle with simple exponential growth and leading indicator covariates. Through targeted fine-tuning, we demonstrate that Chronos-2 appears to lack the capacity to perform well on all these tasks simultaneously. Finally, a practical study on real-world data confirms that these failures to exploit leading indicator information persist in the wild. Our results suggest that current foundation models prioritize high-dimensional pattern matching over fundamental temporal reasoning.

---

[1]Layer6 AI. Correspondence to: Nafiseh Ghoroghchian <nafiseh@layer6.ai>, Geroge Stein <george@layer6.ai>.

*Proceedings of the $43^{rd}$ International Conference on Machine Learning*, Seoul, South Korea. PMLR 306, 2026. Copyright 2026 by the author(s).

[1]We release SimpleTimeBench in GIFT-Eval format and our experimental code at https://github.com/layer6ai-labs/SimpleTimeBench.

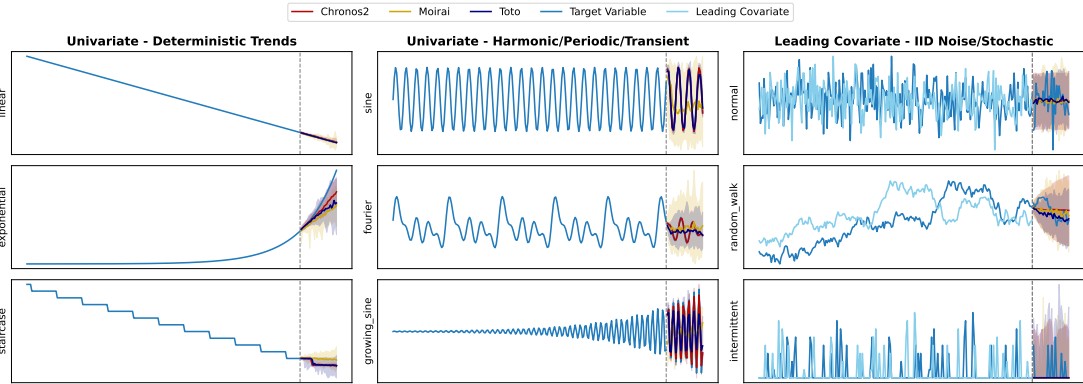

*Figure 1.* Examples of TSFM forecasts on a subset of SimpleTimeBench datasets. Surprising failures include the exponential dataset, and Leading Covariate tests where TSFMs perform *no better than random* even with a perfect leading indicator covariate.

**Related Work**    Synthetic data is ubiquitous in TSFMs (Liu et al., 2025). Approaches generally focus on constructing highly complex and "realistic" mixtures to capture intricate temporal structures and nonstationary regime shifts. (Liu et al., 2025; Wang et al., 2026). However, this emphasis on complex, non-stationary signals often obscures whether models have mastered fundamental temporal logic. Recent diagnostics have begun probing TSFM mechanics and failure modes. For instance, Karaouli et al. (2025) highlight zero-shot generalization limits on synthetic composite waves, while other works identify implicit mean-reverting biases (Yu et al., 2025) and latent representational redundancies (Wiliński et al., 2024). Notably, Chronos (Fatir Ansari et al., 2024) acknowledges inherent difficulties in forecasting unbounded trends due to its specific scaling and normalization logic. Despite these insights, systematic "unit testing" of foundational logic in a broad suite of univariate, multivariate, and covariate-supported regimes remains unexplored. While architectures like Chronos-2 accommodate future-known covariates, there is a critical gap regarding their ability to effectively exploit purely past informative variables, such as deterministic leading indicators.

## 2. SimpleTimeBench: Evaluating Zero-Shot Temporal Logic

We introduce SimpleTimeBench, a diagnostic suite of unit tests designed to isolate foundational forecasting capabilities. SimpleTimeBench generates in three different modes, each meant to test a models' capabilities in different variable dependency settings.

- **Univariate:** Can be used to test the model's ability to extrapolate basic temporal patterns from single-channel time series. These include deterministic trends (e.g., linear, exponential), periodic signals (e.g., sine waves), and various noise processes.

- **Multivariate:** Can be used to evaluate whether adding independent series from the same generative process degrades or improves performance of the model. Each dataset contains multiple ($n = 2$ by default) independent variates drawn from the base distribution, testing if models exhibit dimensional robustness or become distracted by additional channels.

- **Leading Covariate:** Tests whether models can exploit temporal leads by providing covariates that are exact copies of the target series shifted backward in time. This simulates ideal leading indicators where covariates deterministically reveals the target's future. Failure here indicates an inability to utilize predictive covariate information even when the relationship is perfectly informative.

Each mode spans 28 generative processes organized into five regimes detailed in Table 1, where datasets consist of independent series drawn from the distribution.

To quantify forecast topology beyond standard error, we specify a suite of interpretable diagnostic metrics for use with SimpleTimeBench (full details in Appendix A):

- **Relative MAE:** Improvement over naive baselines, representing more basic and interpretable strategies than MASE ($< 1.0$ indicates improvement).

- **Dynamics (MAR, PVR):** Compares dynamic characteristics of predicted time series to ground truth ($> 1$ implies dampened signal; $< 1$ implies exaggeration).

- **Bias (MBR):** Frequency of correct directional alignment relative to historical mean (0.5 is ideal).

This sanitized evaluation can reveal zero-shot blind spots, as demonstrated in our benchmarking of Chronos-2, Moirai and Toto, selected due to their native support for covariate-supported tasks, meaning they can perform all tests in SimpleTimeBench. Selected results are visualized in Figure 1.

*Table 1.* Taxonomy of Time Series Generation Regimes.

| Regime | Description | Prediction Difficulty | Distributions | # |
|---|---|---|---|---|
| **Deterministic** | Low-dimensional, non-repeating growth or states. | **Trivial**: Solvable via simple regression. | `constant`, `linear`, `staircase`, `exponential` | 4 |
| **Periodic** | Fixed-frequency oscillations and synthetic patches. | **Low**: Phase and amplitude estimation. | `sine`, `sawtooth`, `square`, `triangle`, `noise_patch`, `fourier` | 6 |
| **Evolving Periodic** | Signals with time-varying frequency or amplitude. | **Moderate**: Tracks rate-of-change parameters. | `damped_sine`, `growing_sine`, `chirp` | 3 |
| **I.I.D. White Noise** | Independent samples from a fixed distribution. | **Impossible**: Optimal forecast collapses to mean/median. | `normal`, `uniform`, `poisson`, `binary`, `student_t`, `lognormal`, `laplace`, `cauchy`, `skew_normal` | 9 |
| **Stochastic Correlated** | Stochastic processes that are not independent over time, including cumulative processes or sudden impulses. | **Impossible**: Path-dependent; no mean-reversion. | `random_walk`, `piecewise_constant`, `gbm`, `intermittent`, `impulse`, `logistic` | 6 |

While Chronos-2 exhibits the most robust overall performance and successfully forecasts for many datasets, we observe distinct failure modes (FM) across the benchmark shared between all models:

- **FM1: Mean-Reversion Bias** - Across Trend and Periodic tasks, models hedge toward the historical mean rather than tracking amplitude. Average MBR values (0.5 is ideal) of 0.69 (Chronos-2), 0.66 (Toto), and 0.80 (Moirai) on trend & periodic tasks confirm this conservative bias.
- **FM2: Anti-Exponential Bias** - Models struggle with exponential growth. High average Path Volatility Ratios (1.0 is ideal) of 1.90 (Chronos-2), 1.88 (Toto), and 2.20 (Moirai) on trend and period tasks reveal a distinct "smoothing" effect that under-reacts to exponential change.
- **FM3: Covariate Underutilization** - Models systematically fail to exploit leading indicators. Leading covariates act as perfect predictors by providing a time-shifted copy of the target (as shown in the last column of Figure 1). However, on stochastic data, providing this perfect leading indicator yield negligible gain (e.g., Chronos-2 $relMAE_{last}$ shifts only from $1.02_{uni} \rightarrow 0.98_{lead. cov.}$ on random walk, noting lower relMAE is better), suggesting that the models effectively ignore the covariate channel entirely.

## 3. Nature vs. Nurture: Addressing the Zero-Shot Gap via Fine-tuning

Having identified systematic failure modes in TSFMs, we investigate whether these are fundamental architectural limitations or artifacts of pre-training data distribution. Through targeted fine-tuning of Chronos-2[2] , we assess if these behaviours can be corrected:

*Can fine-tuning resolve bias in deterministic trends? (FM1, FM2)* We fine-tuned Chronos-2 on the deterministic trends subset. We observe that the model retains high performance on the constant, linear, and staircase trends, while the MBR improves by 26% for the exponential family.

*Does input normalization inhibit the modeling of unbounded growth? (FM2)* A critical question is whether normalization, widely adopted as a preprocessing step in TSFMs, renders the modeling of exponential growth impossible, where the normalization "squashes" the trend features such that the exponential signal is lost or hard to invert on the target side. To test this, we fine-tuned the model on univariate exponential data. The results demonstrate $> 50\%$ improvement in MAPE, MAR, and PVR for exponential datasets, generalizing to both multivariate and leading covariate settings.

*Is covariate-underutilization an architectural or behavioral constraint? (FM3)* To investigate if the model is architecturally capable of learning from an informative covariate or if the architecture forces a more complex transformation, we fine-tuned the model on leading covariate datasets consisting of I.I.D. White Noise and Stochastic Correlated processes. The results confirm that the architecture **can** support this behavior, with respectively a 17%, 24%, and 24% boost in MAPE, $relMAE_{mean}$, and $relMAE_{last}$.

*Do "All-in-One" foundation models have a capacity bottleneck? (FM1, FM2, FM3)* Fine-tuning on the full SimpleTimeBench yields only marginal gains in overall MAPE, $relMAE_{mean}$, and $relMAE_{last}$ (Table 2), with the model still failing to effectively leverage informative covariates. Instead of developing balanced capabilities across all regimes, the model favors specific tasks at the expense of others. While extended fine-tuning or alternative optimization strategies might mitigate this issue, the simplicity of the underlying data and sufficiency of compute, suggests a fundamental architectural bottleneck in internalizing diverse temporal primitives simultaneously.

---

[2]We thank the Chronos-2 team for releasing their fine-tuning framework, which made this study possible. We focus on Chronos-2 as one of the most capable publicly accessible TSFMs.

See Appendix B for a comprehensive description of the experimental setup and detailed results.

## 4. From Synthetic to Real-World Impact

To investigate covariate underutilization in real world data, we evaluate leading multivariate TSFMs on GIFT-Eval datasets, comparing performance in multivariate mode (using past covariates) versus univariate mode. Table 3 summarizes the results (more details in Appendix C), which show that state-of-the-art multivariate TSFMs often derive minimal benefit from covariates. Chronos-2 and Moirai achieve win rates barely above 50%, scarcely better than random chance, while even ToTo's consistent wins come with negligible margins (1.2% MASE and 1.5% CRPS improvement). Two factors could be driving this: either GIFT-Eval lacks tasks where covariates *are* significantly helpful, or current multivariate models lack ability to extract highly useful information from covariates.

*Table 3.* GIFT-Eval: Multivariate vs. Univariate.

| Model | MASE (vs uni.) | Win Rate | CRPS (vs uni.) | Win Rate |
|---|---|---|---|---|
| Chronos-2 | 0.77 (0.81) | 62.5% | 0.59 (0.62) | 56.3% |
| Moirai | 0.93 (0.93) | 59.1% | 0.75 (0.74) | 50.0% |
| ToTo | 0.72 (0.73) | 100% | 0.65 (0.66) | 87.5% |

To evaluate TSFMs on real-world tasks where covariates are highly predictive, we compared Chronos-2 against a DLinear baseline (Zeng et al., 2023) on two distinct datasets. First, we used hourly streamflow data from two locations on the Delaware River (Hodson et al., 2023), separated by 100 miles with a roughly 14-hour lag in water levels. Second, we designed a synthetic yet illustrative task using historical NVIDIA stock prices to test whether models could forecast accurately *when explicitly provided with future stock price as a leading indicator covariate*. As shown in Figure 2, DLinear, a model with 840 parameters trained for a few minutes on a CPU, successfully exploited these signals, outperforming Chronos-2 when given access to leading indicator information. On the Delaware River dataset, DLinear achieved a 24.2% improvement in MASE with covariate access (8.82 to 6.69), whereas Chronos-2 showed a much smaller gain of 5.9% (7.16 to 6.74). The gap widened in the NVIDIA test: DLinear improved by 62.0% (11.29 to 4.29) compared to only 6.3% for Chronos-2 (10.56 to 9.90). These results are further detailed in Appendix D, illustrating that aggregated evaluation metrics of public benchmarks mask Chronos-2's inability to react to leading information in the covariate. This further suggests that the model struggles to leverage predictive signals in the covariate channel, relying instead on historical target patterns.

## 5. Discussion

While TSFMs demonstrate impressive zero-shot capabilities in many settings, our analysis reveals clear failure modes that warrant caution for industry practitioners in high-stakes environments. These findings suggest that current pre-training regimes might lack the necessary inductive biases to leverage exogenous leading indicators, often defaulting to univariate overreliance despite available predictive signals. To bridge this gap, future research should prioritize refined synthetic data generation that captures a broader diversity of fundamental primitives and multivariate benchmarks with explicit lead-lag relationships. Addressing architectural and training data limitations is essential for TSFMs to move beyond simplistic inference behaviours and become robust, general-purpose forecasting engines capable of navigating complex, real-world dependencies.

## References

Aksu, T., Woo, G., Liu, J., Liu, X., Liu, C., Savarese, S., Xiong, C., and Sahoo, D. GIFT-Eval: A benchmark for general time series forecasting model evaluation. *arXiv preprint arXiv:2410.10393*, 2024.

Ansari, A. F., Shchur, O., Küken, J., Auer, A., Han, B., Mercado, P., Rangapuram, S. S., Shen, H., Stella, L., Zhang, X., et al. Chronos-2: From univariate to universal forecasting. *arXiv preprint arXiv:2510.15821*, 2025.

Cohen, B., Khwaja, E., Wang, K., Masson, C., Ramé, E., Doubli, Y., and Abou-Amal, O. Toto: Time series optimized transformer for observability. *arXiv preprint arXiv:2407.07874*, 2024.

Dziri, N., Lu, X., Sclar, M., Li, X. L., Jiang, L., Lin, B. Y., Welleck, S., West, P., Bhagavatula, C., Le Bras, R., et al. Faith and fate: Limits of transformers on compositionality. *Advances in Neural Information Processing Systems*, 36: 70293–70332, 2023.

Fatir Ansari, A., Stella, L., Turkmen, C., Zhang, X., Mercado, P., Shen, H., Shchur, O., Sundar Rangapuram, S., Pineda Arango, S., Kapoor, S., Zschiegner, J., Maddix, D. C., Wang, H., Mahoney, M. W., Torkkola, K., Wilson, A. G., Bohlke-Schneider, M., and Wang, Y. Chronos: Learning the Language of Time Series. *arXiv e-prints*, art. arXiv:2403.07815, March 2024. doi: 10.48550/arXiv.2403.07815.

Garza, A. and Rosillo, R. TimeCopilot. *arXiv preprint arXiv:2509.00616*, 2025.

Hodson, T. O. et al. dataretrieval: An R and Python package for accessing water data from the U.S. Geological Survey. *Journal of Open Source Software*, 2023.

| | MAPE | | | relMAE$_{mean}$ | | | relMAE$_{last}$ | | |
|---|---|---|---|---|---|---|---|---|---|
| | Base | Finetuned | Improvement | Base | Finetuned | Improvement | Base | Finetuned | Improvement |
| Univariate | 0.57 | 0.51 | 4.14% | 0.54 | 0.43 | 1.70% | 0.61 | 0.61 | 1.70% |
| Multivariate | 0.56 | 0.49 | 4.51% | 0.52 | 0.44 | 1.20% | 0.67 | 0.61 | 1.20% |
| Leading-Covariate | 0.65 | 0.57 | 6.86% | 0.50 | 0.41 | 1.73% | 0.57 | 0.57 | 1.73% |
| Median | 0.58 | 0.52 | **4.62%** | 0.52 | 0.42 | **1.38%** | 0.63 | 0.61 | **1.38%** |

*Table 2.* Performance improvements obtained by fine-tuning Chronos-2 in the *All-in-One* experiment. Each column is first computed separately for each distribution and then aggregated; therefore, the Improvement % column is not simply the difference between the preceding columns.

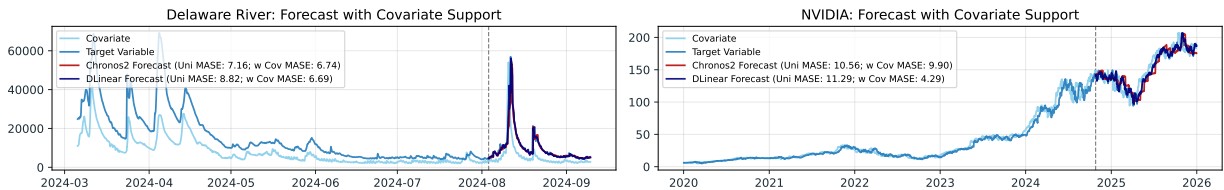

*Figure 2.* Comparison of Chronos-2 vs. DLinear with covariate support on real-world datasets. Forecasts use an expanding-window strategy with horizons of $H = 14$ (Delaware River) and $H = 16$ (NVIDIA). While global performance looks similar, aggregate scales mask Chronos-2's failure to leverage the covariate channel. Granular, zoomed-in comparisons of univariate vs. covariate-support modes demonstrating localized impact are provided in Figure 14 in Appendix D.

Karaouli, N., Coquenet, D., Fromont, E., Mermillod, M., and Reyboz, M. How foundational are foundation models for time series forecasting? *arXiv preprint arXiv:2510.00742*, 2025.

Liu, X., Aksu, T., Liu, J., Wen, Q., Liang, Y., Xiong, C., Savarese, S., Sahoo, D., Li, J., and Liu, C. Empowering Time Series Analysis with Synthetic Data: A Survey and Outlook in the Era of Foundation Models. *arXiv e-prints*, art. arXiv:2503.11411, March 2025. doi: 10.48550/arXiv.2503.11411.

Liu, Y., Qin, G., Shi, Z., Chen, Z., Yang, C., Huang, X., Wang, J., and Long, M. Sundial: A family of highly capable time series foundation models. *arXiv preprint arXiv:2502.00816*, 2025.

Nezhurina, M., Cipolina-Kun, L., Cherti, M., and Jitsev, J. Alice in Wonderland: Simple tasks showing complete reasoning breakdown in state-of-the-art large language models, 2025. URL https://arxiv.org/abs/2406.02061.

Shchur, O., Ansari, A. F., Turkmen, C., Stella, L., Erickson, N., Guerron, P., Bohlke-Schneider, M., and Wang, Y. fev-bench: A realistic benchmark for time series forecasting. *arXiv preprint arXiv:2509.26468*, 2025.

U.S. Geological Survey. National water information system data, 2024. URL https://doi.org/10.5066/F7P55KJN. Gauges 01438500 (Montague) and 01463500 (Trenton).

Wang, W., Wu, K., Wang, D., Zhang, X., et al. Synthetic series-symbol data generation for time series foundation models. *Advances in Neural Information Processing Systems*, 38:10491–10554, 2026.

Wiliński, M., Goswami, M., Potosnak, W., Żukowska, N., and Dubrawski, A. Exploring representations and interventions in time series foundation models. *arXiv preprint arXiv:2409.12915*, 2024.

Woo, G., Liu, C., Kumar, A., Xiong, C., Savarese, S., and Sahoo, D. Unified training of universal time series forecasting transformers. 2024.

Yu, A., Maddix, D. C., Han, B., Zhang, X., Ansari, A. F., Shchur, O., Faloutsos, C., Wilson, A. G., Mahoney, M. W., and Wang, Y. Understanding the implicit biases of design choices for time series foundation models. *arXiv preprint arXiv:2510.19236*, 2025.

Zeng, A., Chen, M., Zhang, L., and Xu, Q. Are transformers effective for time series forecasting? In *Proceedings of the AAAI Conference on Artificial Intelligence*, 2023.

# A. SimpleTimeBench

SimpleTimeBench is a diagnostic benchmark suite designed to evaluate time series foundation models on fundamental forecasting primitives through controlled synthetic tasks and appropriate metrics to measure their responses. Unlike traditional benchmarks that emphasize distributional diversity, SimpleTimeBench focuses on *interpretable failure modes* by testing whether models can handle scenarios where the solution is transparent and should be trivial for classical methods.

Rather than a "leaderboard" style benchmark, SimpleTimeBench is meant to help model developers and industry practitioners probe and evaluate characteristics of models, in the ultimate goal that the community can further push the state-of-the-art in time series foundation modeling. The underlying premise of our work is that a truly foundational model should be able to perform reasonably on a wide range of time series tasks, from the most simple (extrapolate a univariate constant line) to complex real world multivariate tasks, and everything in between. Here we focus on simple fundamental time series components in order to isolate any potential unwanted effects from noisy and complex data.

## A.1. Datasets

SimpleTimeBench is built to generate series from 28 distinct generative processes, meant to capture time series primitives that can appear in forecasting tasks. Series sampled from each process are intended to have known solutions for what an "ideal model" is expected to output as its forecast. In this manner, forecasts from a given model can be directly compared to the ideal solution, rather than using a real or complex synthetic dataset where the performance floor of the model is often unknown. SimpleTimeBench time series are meant to be "solvable". This can either mean exactly predicting future values, or in the case where future values are not predictable from past values, the model recognizing this and predicting an appropriate distribution. For example, for a time series drawn from the IID normal distribution, the ideal model predicts the mean of the data ($\mu$) and acknowledges the constant noise level ($\sigma$), while for a series drawn from a random walk distribution, an ideal model should predict the last value with an uncertainty that grows in proportion over time proportional to the square root of steps.

Series from a given process are contained within a dataset, and datasets are grouped into the following five generative regimes:

**Regime Descriptions:**

- **Deterministic Trends:** Low-dimensional patterns solvable via parametric regression.

- **Periodic:** Fixed-frequency oscillations predictable through phase and amplitude estimation.

- **Evolving Periodic:** Signals with time-varying frequency or amplitude that require adaptive parameter tracking.

- **I.I.D. White Noise:** Independent samples from a continuous probability distributions.

- **Stochastic Correlated:** Non-stationary processes with correlated randomness.

Table 1 lists the distributions included in each generative regime, together with an assumed difficulty of the forecasting problem.

### A.1.1. TASK TYPES

Figures 3, 4, 5 display samples from all distributions for each task type describe in Section 2: univariate, multivariate, and leading covariate. Note that we choose 48 steps as the default lag for the leading covariate.

## A.2. Metrics

To surface failure modes beyond aggregate point forecast accuracy, we introduce a suite of diagnostic metrics that characterize *how* models fail. While traditional metrics like CRPS measure aggregate error, they cannot distinguish between systematic biases, lack of path volatility, or hedging toward historical means.

The metrics presented in Table 4 and defined below are designed to detect specific pathologies in time series forecasts, and are referenced to a straightforward *reference value* to diagnose forecast characteristics. Therefore, they are generally **not** meant to be used as leaderboard targets, as individual metrics do not give full picture of the model's forecast accuracy.

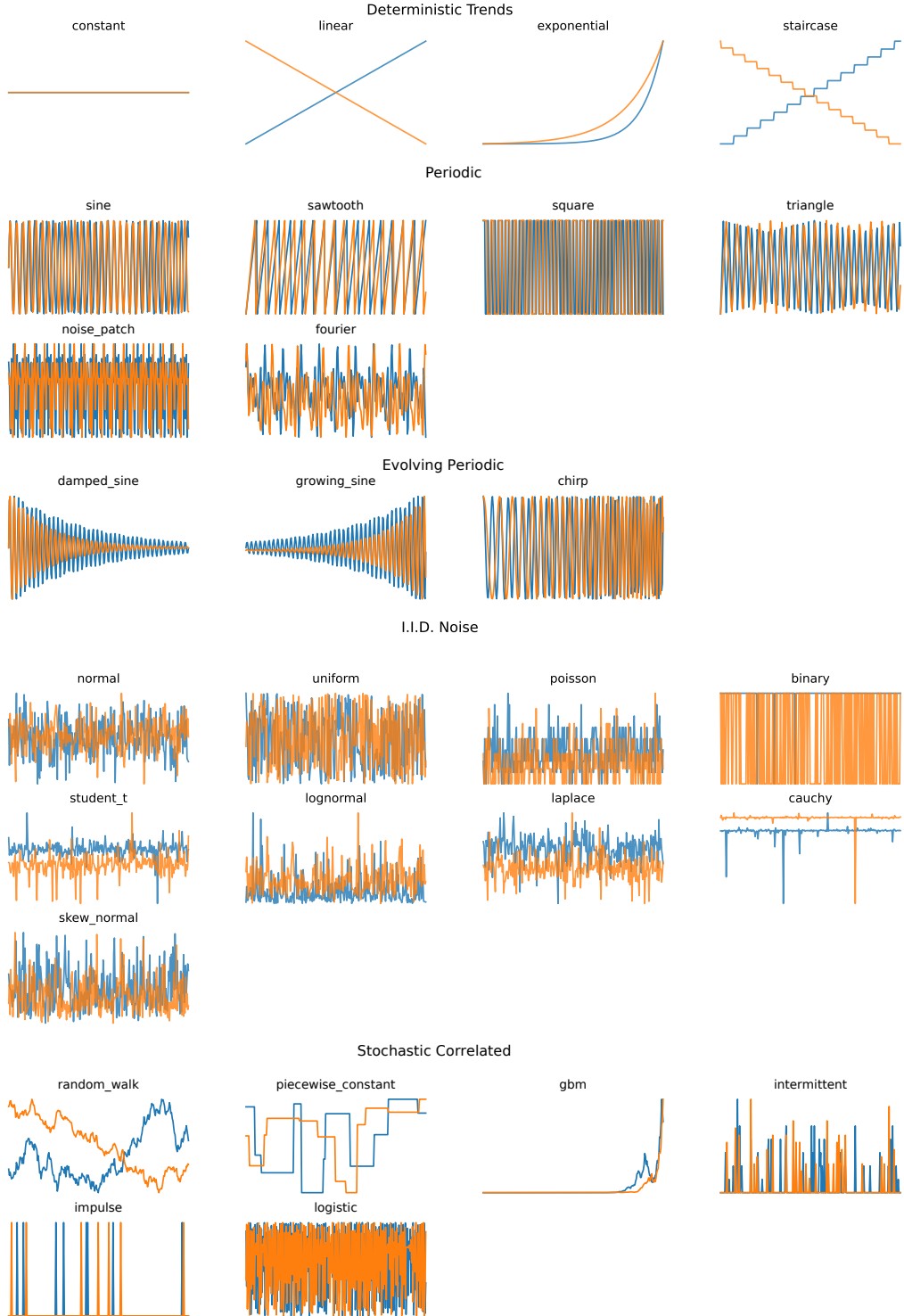

*Figure 3.* Samples from all SimpleTimeBench distributions in Univariate generation mode. Colors represent different series. Series are min-max normalized for easier visualization. Two series are shown in each panel.

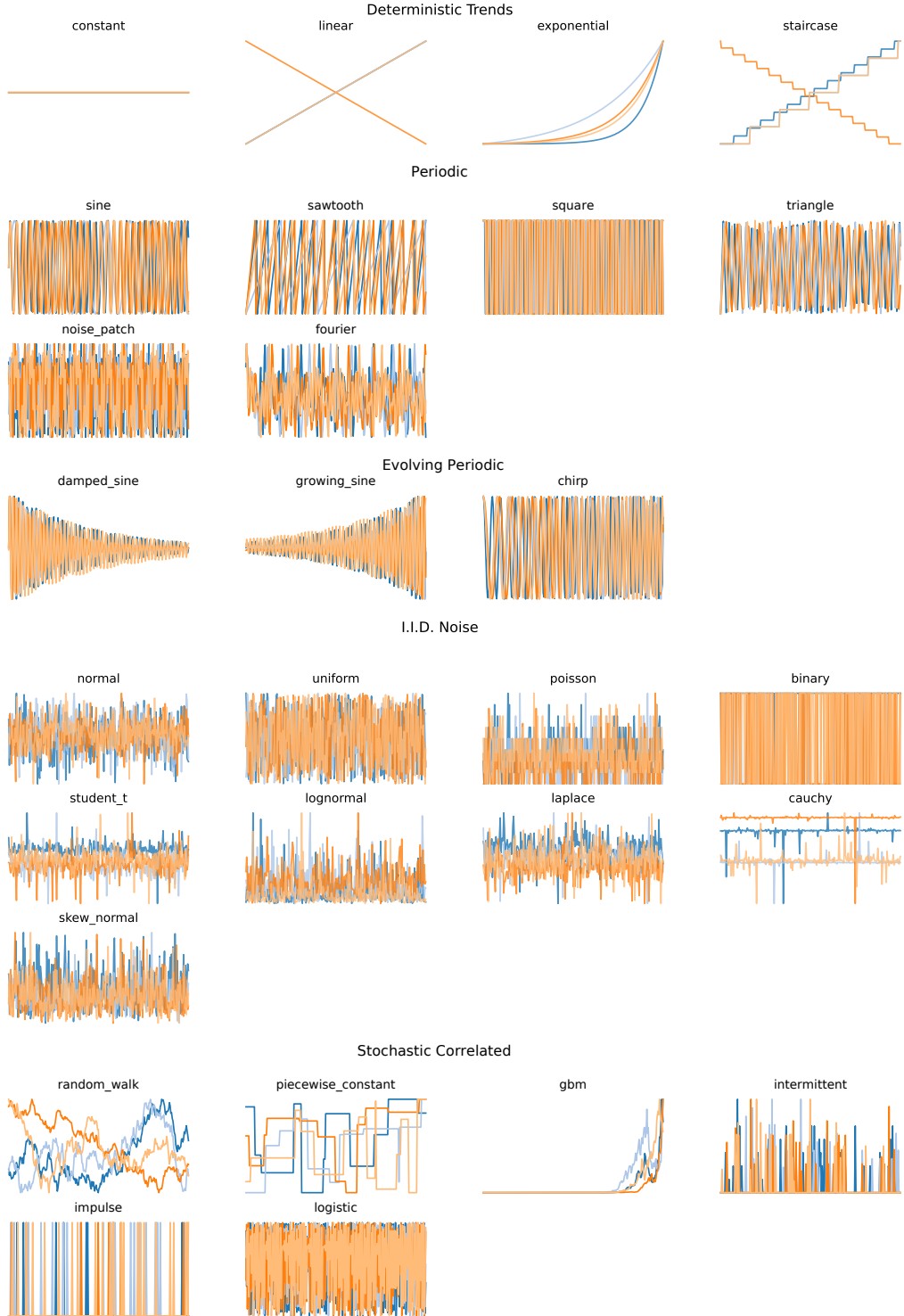

*Figure 4.* Samples from all SimpleTimeBench distributions in Multivariate generation mode. Colors represent different series, shades represent covariates. Series are min-max normalized for easier visualization. Two series are shown in each panel.

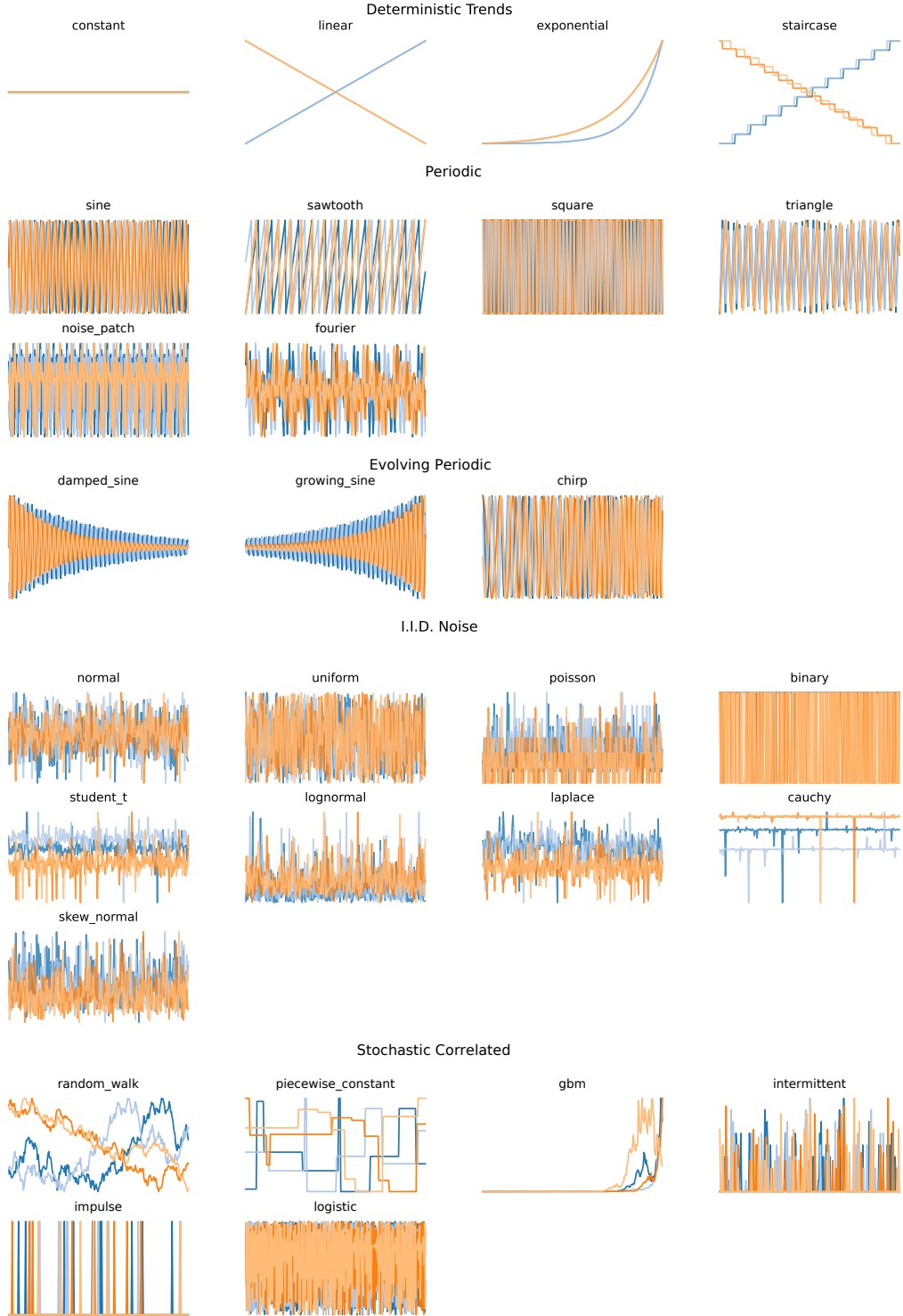

*Figure 5.* Samples from all SimpleTimeBench distributions in Leading Covariate generation mode. Colors represent different series, shades represent covariates, where lighter shades (leading covariates) lead darker (main variate). Series are min-max normalized for easier visualization. Two series are shown in each panel.

*Table 4.* Time Series Forecasting Metrics: Interpretation Guide with Noise-Aware Bounds

| Metric Name (Abbreviation) | What it Measures | Formula Logic | When to use | Ref. | < Ref. Value (Min Bound) | > Ref. Value (Max Bound) |
|---|---|---|---|---|---|---|
| **Relative MAEs** (relMAE$_{\text{mean}}$, relMAE$_{\text{last}}$) | **Skill.** Compares model error vs a naive baseline. | Ratio of Average Model Error to Average Naive Error. | **Benchmarking.** To verify the model has learned anything useful. | 1.0 | **Good** $[0, 1.0)$: Beating the baseline; extracting signal. | **Bad** $[1.0, \infty)$: Worse than a naive flat-line guess. |
| **Mean Bias Ratio** (MBR) | **Hedging.** Frequency of predictions that are biased towards the mean. | Frequency of model landing between the historical mean and the target. | **Behavior.** To see if the model is "playing it safe." | 0.5 | **Aggressive** $[0, 0.5)$: Model frequently overshoots the trend. | **Hedging** $(0.5, 1.0)$: Model tends towards the mean to avoid penalty. |
| **Mean Attraction Ratio** (MAR) | **Dynamics.** Ratio of target's deviation from mean vs forecast's deviation. | Average deviation of target from mean divided by average deviation of forecast from mean. | **Volatility.** When catching extreme events (spikes) is critical. | 1.0 | **Hyper-Active** $[0, 1.0)$: Model exaggerates peaks & valleys. | **Muted** $(1.0, \infty)$: Model dampens signal; under-sizes moves. |
| **Path Volatility Ratio** (PVR) | **Texture.** High frequency characteristics of target vs forecast. | Sum of absolute differences in target path divided by same for forecast path. | **Realism.** To ensure the curve shape/noise looks realistic. | 1.0 | **Hallucinating** $[0, 1.0)$: Adds jitter or noise not in reality. | **Too Smooth** $(1.0, \infty)$: Misses rapid changes; overly smooth. |

*The noise floor is data-dependent. For deterministic signals, the reference value is $\approx 0$.

**relMAE$_{\text{last}}$:** Measures whether the model outperforms a trivial baseline that simply repeats the last observed value for all future time steps:

$$\text{relMAE}_{\text{last}} = \frac{\text{MAE}_{\text{model}}}{\text{MAE}_{\text{persistence}} + \epsilon}$$

where $\text{MAE}_{\text{persistence}}$ uses the last observation as the forecast, and $\epsilon$ is a small constant to avoid division by zero. Values $> 1$ indicate the model performs worse than this naive baseline.

**relMAE$_{\text{mean}}$:** Measures whether the model outperforms predicting the historical mean for all future time steps:

$$\text{relMAE}_{\text{mean}} = \frac{\text{MAE}_{\text{model}}}{\text{MAE}_{\text{mean}} + \epsilon}$$

where $\text{MAE}_{\text{mean}}$ uses the historical mean as the forecast. Values $> 1$ indicate the model performs worse than this simple baseline.

**Mean Attraction Ratio (MAR):** Quantifies whether the forecast matches the global amplitude of the target by comparing average deviations from the historical mean:

$$\text{MAR} = \frac{\mathbb{E}[|y_t - \bar{y}_{\text{hist}}|]}{\mathbb{E}[|\hat{y}_t - \bar{y}_{\text{hist}}|] + \epsilon}$$

Values $> 1$ indicate forecasts are "muted" (under-sizing moves), while values $< 1$ indicate over-sizing.

**Path Volatility Ratio (PVR):** Measures whether the forecast path exhibits the correct high-frequency characteristics by comparing first differences:

$$\text{PVR} = \frac{\mathbb{E}[|y_t - y_{t-1}|]}{\mathbb{E}[|\hat{y}_t - \hat{y}_{t-1}|] + \epsilon}$$

Values $> 1$ indicate forecasts are smoother than the true signal, while values $< 1$ indicate excessive jitter.

**Mean Bias Ratio (MBR):** Quantifies how often the model hedges toward the historical mean rather than committing to the target direction:

$$\text{MBR} = \mathbb{P}(\text{sign}(y_t - \bar{y}_{\text{hist}}) = \text{sign}(y_t - \hat{y}_t))$$

Values near 0.5 are ideal (forecast is on the correct side of the mean as often as the target). Values $> 0.7$ suggest systematic mean-hedging behavior.

**Note:** In all metric definitions, $\epsilon$ is a small constant (e.g., $10^{-10}$) added to prevent division by zero.

### A.3. Results

For the results presented in this work we focus on univariate forecasting, and forecasting with past covariates in both Leading and non-Leading modes. Forecast accuracy is only measured for the target variable (var_0). We leave multivariate forecasting and future covariates for future work, noting that the datasets here can be used for those tasks as well.

Each dataset defaults to 1024 samples with a default length of 256 time steps, with a forecast horizon of 48 and a context length of 208. This is the setting used for all experiments in this paper, but an initial investigation shows characteristics of TSFMs discussed above occur at a wide variety of parameter settings. TSFMs should be able to solve these tasks at a wide variety of context lengths, up to a minimum where there is insufficient historical window to observe; we leave such an investigation of the effects of context length for future work. The benchmark datasets are in GIFT-Eval-compatible format to facilitate standardized evaluation.

We evaluate Chronos-2, Toto, and Moirai, due to their native support for covariate-supported tasks, meaning they can perform all tests with past covariates support and popularity. Figure 6 displays example forecasts for the models on each dataset.

Tables 5 and 6 compare the performance of Chronos-2, Toto, and Moirai on each subset of the SimpleTimeBench using selected simpletimebench metrics.

For regimes where TSFMs should get good predictions we display Mean Attraction Ratio (MAR), Mean Bias Ratio (MBR), Path Volatility Ratio (PVR), and Relative MAEs ($relMAE_{mean}$, and $relMAE_{last}$). All metrics are measured over the target variable - here the univariate, past covariate, and past leading covariate modes are meant to test whether covariate information (unrelated to the target forecast but "in context" for past covatiate) affects the models ability to forecast. Interestingly we find that past (unrelated) covariate information can actually be harmful for model predictions, see Moirai and Toto $relMAE_{last}$ on linear

For "unpredictable" regimes we examine $relMAE_{mean}$ and $relMAE_{last}$. These we find that adding past *leading* covariate information does not significantly help. We note that as explained above the ideal forecast for some distributions is neither the mean or the last value (persistence), so values $>1$ on those distributions do not necessarily mean model issues.

### A.4. Chronos-2 Future Variate Usage Validation

The above results demonstrate that past covariates are not heavily utilized by TSFMs. Here we perform a simple validation check that a future covariate is, as demonstrated clearly in (Ansari et al., 2025)

To validate Chronos-2's basic usage of future covariates in our experimental setup, we show results in Figure 7 where the model forecasts random walks given either past-only or future covariates that replicate the target signal without any lag. As expected, when given the future values of the target through a future covariate, the model can predict it nearly perfectly. This stands in contrast to the cases in Sections 2 and 4 where the model is instead given future values through a leading past-only covariate, and fails to use this information.

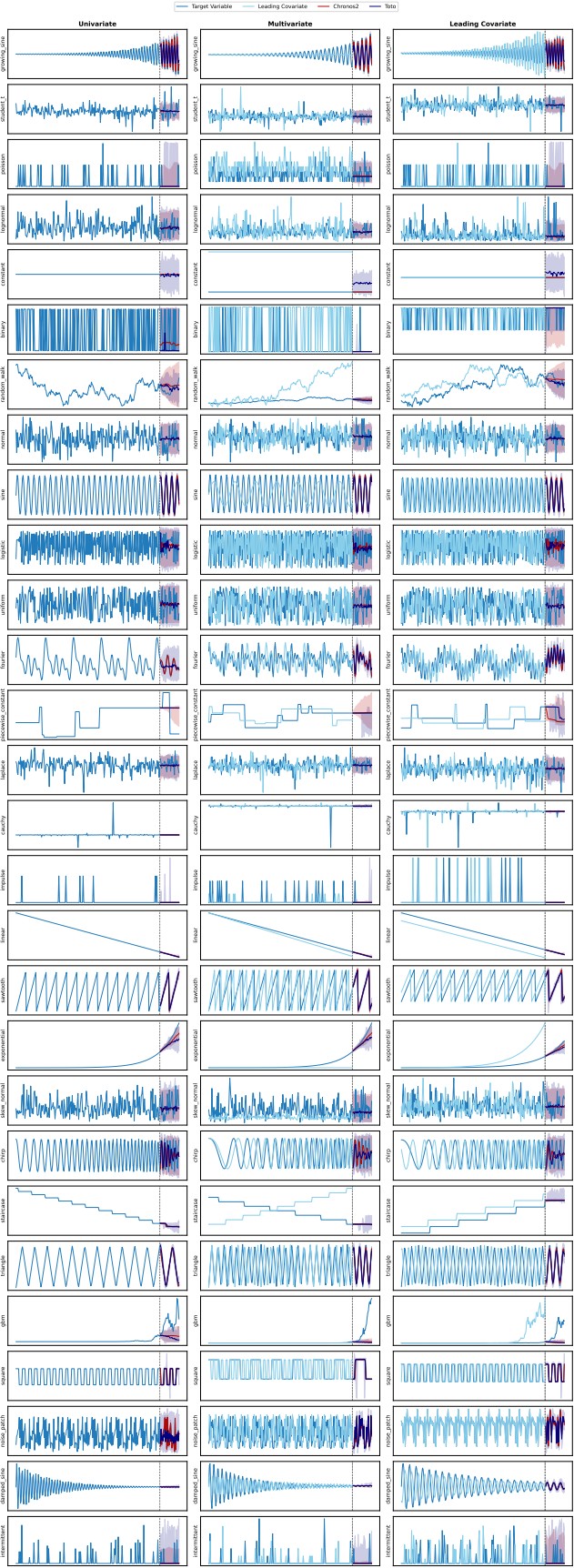

*Figure 6.* Samples of all distributions in SimpleTimeBench.

*Table 5.* SimpleTimeBench Metrics on Deterministic Trends, Periodic, and Evolving Periodic. Metrics are displayed for univariate mode (Univ), past covariates from same distribution (Cov.), and past leading covariates (Lead).

| regime | model | mode dataset | MeanAttractionRatio Univ | Cov | Lead | MeanBiasRatio Univ | Cov | Lead | relMAE$_{mean}$ Univ | Cov | Lead | PathVolatilityRatio Univ | Cov | Lead | relMAE$_{last}$ Univ | Cov | Lead |
|---|---|---|---|---|---|---|---|---|---|---|---|---|---|---|---|---|---|
| Deterministic Trends | Chronos2 | exponential | 1.31 | 1.32 | 1.19 | 0.92 | 0.87 | 0.92 | 0.24 | 0.24 | 0.16 | 1.90 | 1.90 | 1.57 | 0.37 | 0.37 | 0.28 |
| | | linear | 1.01 | 1.01 | 1.01 | 0.98 | 0.97 | 0.98 | 0.01 | 0.01 | 0.01 | 1.03 | 1.03 | 1.03 | 0.03 | 0.04 | 0.03 |
| | | staircase | 1.01 | 1.01 | 1.01 | 0.72 | 0.67 | 0.69 | 0.01 | 0.01 | 0.01 | 0.70 | 0.74 | 0.74 | 0.05 | 0.06 | 0.06 |
| | Moirai | exponential | 1.67 | 1.54 | 1.28 | 0.91 | 0.51 | 0.52 | 0.40 | 1.44 | 1.57 | 2.21 | 1.26 | 0.88 | 0.62 | 2.20 | 2.67 |
| | | linear | 1.01 | 0.53 | 0.47 | 0.53 | 0.75 | 0.76 | 0.02 | 2.37 | 2.66 | 0.82 | 0.15 | 0.13 | 0.10 | 12.39 | 13.88 |
| | | staircase | 1.10 | 1.07 | 0.95 | 0.78 | 0.84 | 0.79 | 0.11 | 1.19 | 1.30 | 0.42 | 0.18 | 0.18 | 0.55 | 5.93 | 6.70 |
| | Toto | exponential | 1.58 | 1.43 | 1.24 | 0.99 | 0.51 | 0.52 | 0.37 | 1.48 | 1.59 | 1.88 | 1.12 | 0.92 | 0.57 | 2.27 | 2.70 |
| | | linear | 1.00 | 0.52 | 0.46 | 0.12 | 0.75 | 0.75 | 0.00 | 2.39 | 2.68 | 0.99 | 0.15 | 0.13 | 0.02 | 12.47 | 14.00 |
| | | staircase | 1.04 | 1.01 | 0.92 | 0.63 | 0.79 | 0.76 | 0.05 | 1.21 | 1.31 | 0.57 | 0.22 | 0.20 | 0.23 | 6.02 | 6.75 |
| Periodic | Chronos2 | damped sine | 1.08 | 1.13 | 1.13 | 0.68 | 0.74 | 0.81 | 0.13 | 0.15 | 0.14 | 1.10 | 1.15 | 1.14 | 0.09 | 0.10 | 0.09 |
| | | fourier | 1.01 | 1.01 | 1.02 | 0.55 | 0.56 | 0.57 | 0.10 | 0.12 | 0.12 | 1.01 | 1.02 | 1.02 | 0.07 | 0.09 | 0.08 |
| | | growing sine | 1.17 | 1.17 | 1.12 | 0.90 | 0.85 | 0.87 | 0.15 | 0.15 | 0.12 | 1.18 | 1.18 | 1.13 | 0.14 | 0.14 | 0.11 |
| | | noise patch | 0.99 | 1.00 | 1.00 | 0.50 | 0.52 | 0.53 | 0.07 | 0.08 | 0.08 | 0.99 | 1.00 | 1.01 | 0.05 | 0.06 | 0.06 |
| | | sawtooth | 1.00 | 1.01 | 1.01 | 0.56 | 0.56 | 0.55 | 0.03 | 0.03 | 0.03 | 1.00 | 1.00 | 1.00 | 0.02 | 0.02 | 0.02 |
| | | sine | 1.01 | 1.01 | 1.01 | 0.63 | 0.63 | 0.63 | 0.03 | 0.05 | 0.04 | 1.01 | 1.02 | 1.02 | 0.03 | 0.04 | 0.03 |
| | | square | 1.01 | 1.02 | 1.02 | 0.54 | 0.67 | 0.70 | 0.04 | 0.04 | 0.04 | 0.99 | 0.99 | 0.99 | 0.04 | 0.04 | 0.04 |
| | | triangle | 1.00 | 1.00 | 1.00 | 0.47 | 0.46 | 0.47 | 0.05 | 0.07 | 0.07 | 1.06 | 1.07 | 1.07 | 0.04 | 0.05 | 0.05 |
| | Moirai | damped sine | 1.19 | 1.25 | 1.21 | 0.78 | 0.75 | 0.78 | 1.20 | 1.37 | 1.46 | 1.28 | 1.45 | 1.39 | 0.81 | 0.93 | 0.99 |
| | | fourier | 2.80 | 3.68 | 3.31 | 0.89 | 0.91 | 0.91 | 1.03 | 1.04 | 1.06 | 2.86 | 3.45 | 3.12 | 0.72 | 0.73 | 0.75 |
| | | growing sine | 4.28 | 5.55 | 4.11 | 0.93 | 0.89 | 0.87 | 0.99 | 1.06 | 1.09 | 6.30 | 6.54 | 5.12 | 0.89 | 0.96 | 0.97 |
| | | noise patch | 4.00 | 2.25 | 2.33 | 0.93 | 0.89 | 0.89 | 0.98 | 1.11 | 1.10 | 5.73 | 6.04 | 5.90 | 0.75 | 0.85 | 0.83 |
| | | sawtooth | 1.52 | 1.89 | 1.36 | 0.79 | 0.87 | 0.82 | 0.99 | 1.12 | 1.24 | 1.61 | 1.63 | 1.53 | 0.74 | 0.83 | 0.92 |
| | | sine | 1.57 | 1.82 | 1.71 | 0.82 | 0.85 | 0.84 | 1.00 | 1.17 | 1.18 | 1.57 | 1.70 | 1.54 | 0.78 | 0.91 | 0.92 |
| | | square | 1.07 | 1.05 | 1.04 | 0.80 | 0.79 | 0.82 | 0.92 | 1.01 | 1.02 | 1.38 | 1.56 | 1.45 | 0.91 | 1.00 | 1.01 |
| | | triangle | 1.83 | 2.06 | 2.11 | 0.84 | 0.88 | 0.88 | 1.03 | 1.12 | 1.11 | 1.97 | 1.97 | 2.02 | 0.77 | 0.83 | 0.82 |
| | Toto | damped sine | 1.03 | 1.03 | 1.03 | 0.55 | 0.72 | 0.75 | 0.33 | 1.50 | 1.54 | 1.09 | 1.08 | 1.08 | 0.22 | 1.02 | 1.05 |
| | | fourier | 1.49 | 1.49 | 1.59 | 0.74 | 0.81 | 0.82 | 0.57 | 1.22 | 1.20 | 1.63 | 1.59 | 1.69 | 0.40 | 0.85 | 0.85 |
| | | growing sine | 2.01 | 2.01 | 1.73 | 0.89 | 0.79 | 0.79 | 0.53 | 1.23 | 1.30 | 2.08 | 2.03 | 1.78 | 0.48 | 1.11 | 1.16 |
| | | noise patch | 1.50 | 1.29 | 1.36 | 0.75 | 0.81 | 0.82 | 0.57 | 1.25 | 1.23 | 1.61 | 1.63 | 1.72 | 0.44 | 0.96 | 0.93 |
| | | sawtooth | 1.01 | 1.01 | 1.02 | 0.53 | 0.75 | 0.75 | 0.08 | 1.29 | 1.30 | 1.04 | 1.01 | 1.01 | 0.06 | 0.96 | 0.97 |
| | | sine | 1.00 | 1.00 | 1.00 | 0.51 | 0.75 | 0.75 | 0.14 | 1.40 | 1.40 | 1.04 | 1.01 | 1.00 | 0.11 | 1.09 | 1.09 |
| | | square | 1.01 | 1.01 | 1.02 | 0.83 | 0.91 | 0.93 | 0.13 | 0.99 | 1.00 | 1.00 | 0.98 | 0.98 | 0.13 | 0.99 | 0.99 |
| | | triangle | 0.94 | 0.95 | 0.95 | 0.40 | 0.74 | 0.74 | 0.20 | 1.35 | 1.35 | 1.12 | 1.08 | 1.08 | 0.15 | 1.00 | 1.00 |
| Evolving Periodic | Chronos2 | chirp | 1.97 | 1.99 | 1.96 | 0.87 | 0.87 | 0.86 | 1.06 | 1.05 | 1.02 | 2.83 | 2.83 | 2.68 | 0.83 | 0.82 | 0.79 |
| | | damped sine | 1.08 | 1.13 | 1.13 | 0.68 | 0.74 | 0.81 | 0.13 | 0.15 | 0.14 | 1.10 | 1.15 | 1.14 | 0.09 | 0.10 | 0.09 |
| | | growing sine | 1.17 | 1.17 | 1.12 | 0.90 | 0.85 | 0.87 | 0.15 | 0.15 | 0.12 | 1.18 | 1.18 | 1.13 | 0.14 | 0.14 | 0.11 |
| | Moirai | chirp | 2.69 | 2.79 | 2.74 | 0.92 | 0.92 | 0.92 | 1.04 | 1.05 | 1.05 | 3.23 | 3.65 | 3.53 | 0.81 | 0.81 | 0.82 |
| | | damped sine | 1.19 | 1.25 | 1.21 | 0.78 | 0.75 | 0.78 | 1.20 | 1.37 | 1.46 | 1.28 | 1.45 | 1.39 | 0.81 | 0.93 | 0.99 |
| | | growing sine | 4.28 | 5.55 | 4.11 | 0.93 | 0.89 | 0.87 | 0.99 | 1.06 | 1.09 | 6.30 | 6.54 | 5.12 | 0.89 | 0.96 | 0.97 |
| | Toto | chirp | 2.10 | 2.20 | 2.04 | 0.88 | 0.90 | 0.89 | 1.09 | 1.09 | 1.10 | 2.68 | 2.65 | 2.49 | 0.85 | 0.85 | 0.86 |
| | | damped sine | 1.03 | 1.03 | 1.03 | 0.55 | 0.72 | 0.75 | 0.33 | 1.50 | 1.54 | 1.09 | 1.08 | 1.08 | 0.22 | 1.02 | 1.05 |
| | | growing sine | 2.01 | 2.01 | 1.73 | 0.89 | 0.79 | 0.79 | 0.53 | 1.23 | 1.30 | 2.08 | 2.03 | 1.78 | 0.48 | 1.11 | 1.16 |

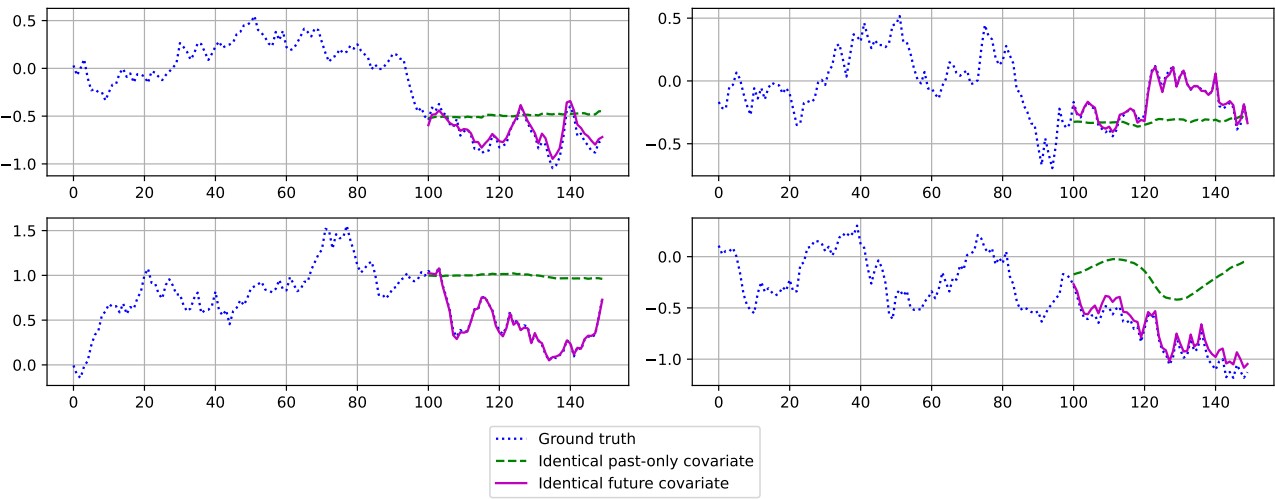

- - - - Ground truth
- - - Identical past-only covariate
——— Identical future covariate

*Figure 7.* Evaluation of Chronos-2 predictions when a sequence identical to the target sequence is given with no lag, either as a past-only or future covariate, using random walks. In the future covariate case, Chronos-2 is able to correctly reproduce the covariate as its prediction.

*Table 6.* SimpleTimeBench Metrics on IID Noise and Stochastic Correlated processes. Metrics are displayed for univariate mode (Univ), past covariates from same distribution (Cov.), and past leading covariates (Lead).

| regime | model | mode dataset | relMAE$_{mean}$ Univ | Cov | Lead | relMAE$_{last}$ Univ | Cov | Lead |
|---|---|---|---|---|---|---|---|---|
| I.I.D. Noise | Chronos2 | binary | 0.78 | 0.78 | 0.78 | 0.80 | 0.80 | 0.79 |
| | | cauchy | 0.69 | 0.69 | 0.69 | 0.65 | 0.65 | 0.65 |
| | | laplace | 1.00 | 1.00 | 1.00 | 0.68 | 0.68 | 0.68 |
| | | lognormal | 0.94 | 0.94 | 0.94 | 0.68 | 0.68 | 0.68 |
| | | normal | 1.00 | 1.00 | 1.00 | 0.71 | 0.71 | 0.70 |
| | | poisson | 0.98 | 0.98 | 0.97 | 0.73 | 0.73 | 0.73 |
| | | skew normal | 0.99 | 0.99 | 0.99 | 0.71 | 0.71 | 0.70 |
| | | student t | 1.01 | 1.00 | 1.00 | 0.69 | 0.68 | 0.68 |
| | | uniform | 1.00 | 1.00 | 1.00 | 0.76 | 0.76 | 0.75 |
| | Moirai | binary | 0.78 | 1.52 | 1.50 | 0.80 | 1.54 | 1.49 |
| | | cauchy | 0.70 | 0.63 | 0.80 | 0.65 | 0.60 | 0.77 |
| | | laplace | 1.02 | 1.02 | 1.02 | 0.69 | 0.70 | 0.70 |
| | | lognormal | 0.95 | 0.96 | 0.96 | 0.69 | 0.70 | 0.70 |
| | | normal | 1.02 | 1.01 | 1.01 | 0.72 | 0.72 | 0.71 |
| | | poisson | 0.99 | 1.51 | 1.54 | 0.74 | 1.14 | 1.15 |
| | | skew normal | 1.01 | 1.32 | 1.32 | 0.72 | 0.95 | 0.94 |
| | | student t | 1.01 | 1.01 | 1.01 | 0.69 | 0.68 | 0.69 |
| | | uniform | 1.02 | 1.02 | 1.02 | 0.77 | 0.77 | 0.77 |
| | Toto | binary | 0.76 | 1.52 | 1.50 | 0.78 | 1.55 | 1.49 |
| | | cauchy | 0.70 | 0.62 | 0.79 | 0.66 | 0.59 | 0.77 |
| | | laplace | 1.00 | 1.01 | 1.01 | 0.68 | 0.69 | 0.68 |
| | | lognormal | 0.96 | 0.96 | 0.97 | 0.69 | 0.70 | 0.70 |
| | | normal | 1.01 | 1.00 | 1.01 | 0.71 | 0.71 | 0.70 |
| | | poisson | 0.97 | 1.46 | 1.48 | 0.72 | 1.10 | 1.11 |
| | | skew normal | 1.01 | 1.27 | 1.27 | 0.72 | 0.91 | 0.91 |
| | | student t | 1.00 | 1.00 | 1.00 | 0.68 | 0.68 | 0.68 |
| | | uniform | 1.01 | 1.01 | 1.01 | 0.76 | 0.76 | 0.76 |
| Stochastic Correlated | Chronos2 | gbm | 0.91 | 0.93 | 0.92 | 0.94 | 0.95 | 0.94 |
| | | impulse | 0.54 | 0.54 | 0.54 | 0.55 | 0.55 | 0.57 |
| | | intermittent | 0.58 | 0.59 | 0.59 | 0.58 | 0.58 | 0.57 |
| | | logistic | 1.02 | 1.00 | 0.97 | 0.80 | 0.78 | 0.76 |
| | | piecewise constant | 0.73 | 0.72 | 0.66 | 1.14 | 1.12 | 1.08 |
| | | random walk | 0.47 | 0.47 | 0.45 | 1.02 | 1.02 | 0.98 |
| | Moirai | gbm | 0.97 | 1.03 | 1.07 | 0.99 | 1.05 | 1.12 |
| | | impulse | 0.54 | 0.53 | 0.54 | 0.55 | 0.55 | 0.54 |
| | | intermittent | 0.58 | 0.58 | 0.58 | 0.57 | 0.59 | 0.59 |
| | | logistic | 1.01 | 1.01 | 1.01 | 0.79 | 0.79 | 0.79 |
| | | piecewise constant | 0.67 | 1.33 | 1.33 | 1.04 | 2.07 | 2.25 |
| | | random walk | 0.49 | 2.25 | 2.36 | 1.07 | 4.75 | 5.06 |
| | Toto | gbm | 0.99 | 1.01 | 1.04 | 1.01 | 1.04 | 1.09 |
| | | impulse | 0.53 | 0.53 | 0.54 | 0.54 | 0.56 | 0.54 |
| | | intermittent | 0.58 | 0.58 | 0.59 | 0.57 | 0.60 | 0.59 |
| | | logistic | 1.01 | 1.03 | 1.03 | 0.79 | 0.81 | 0.81 |
| | | piecewise constant | 0.66 | 1.33 | 1.33 | 1.04 | 2.07 | 2.25 |
| | | random walk | 0.50 | 2.20 | 2.32 | 1.09 | 4.63 | 4.96 |

## B. Fine-tuning Experiments

Our fine-tuning framework uses the full fine-tuning recipe from the Chronos-2 GitHub repository[3].

### B.1. Experimental Design and Research Questions

Our experimental framework is structured around four distinct tests designed to address the foundational questions posed in Sec. 3 regarding model robustness and adaptability.

- *Bias Experiment* (Can foundation models maintain zero-shot linear and non-linear trajectories?): We evaluate the models' inherent inductive biases by testing their ability to continue simple constant, linear, exponential, and staircase trends. Using 100,000 samples per distribution with a 10% validation split for fine-tuning, we measure whether the models can transcend "mean-reversion" tendencies and maintain deterministic slopes over long horizons. Fig. 8 demonstrates clear performance improvement in deterministic trends after fine-tuning.

---

[3] https://github.com/amazon-science/chronos-forecasting

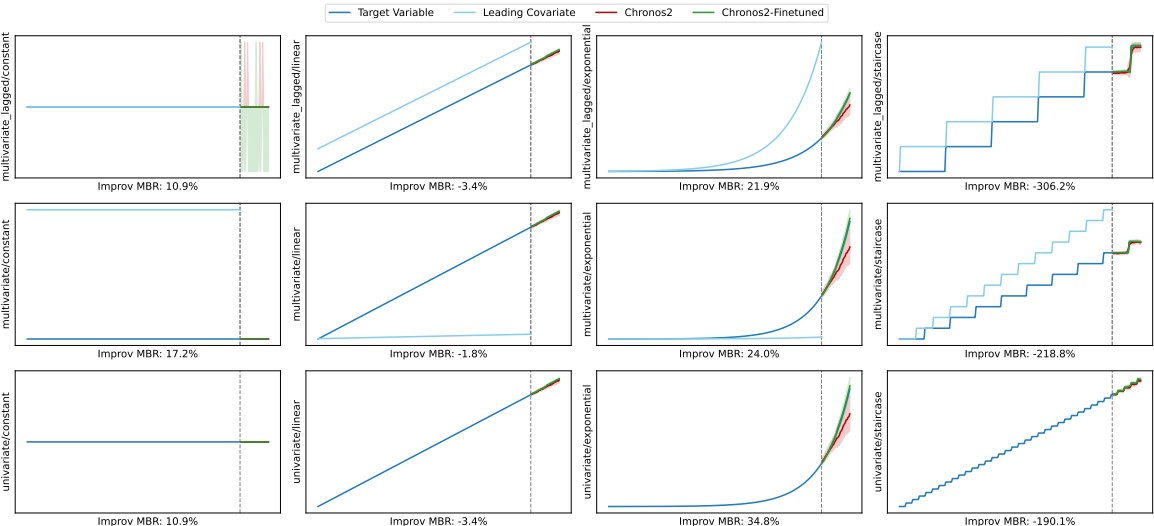

*Figure 8.* Performance of Chronos-2 model on Deterministic Trends after fine-tuning in the *Bias* experiment. The values in each cell represent the percentage improvement of the corresponding metric calculated over 100 samples from that distribution.

- *Norm Experiment*: (To what extent do models capture evolving periodicity and phase consistency?) This experiment focuses on univariate exponential trend. Here, we use 100,000 samples of univariate exponential distribution with a $10\%$ validation split for fine-tuning. Fig. 9 clearly show how exponential trends could be perfectly predicted after fine-tuning.

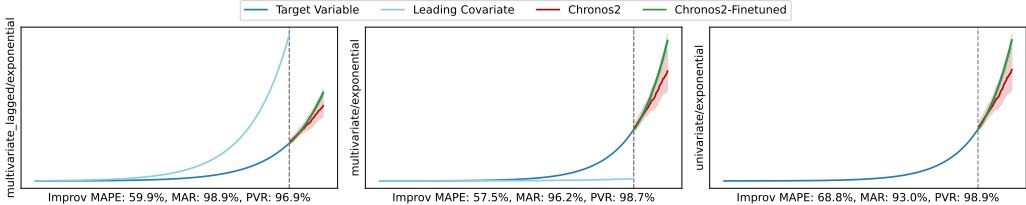

*Figure 9.* Performance of Chronos-2 model on exponential trends after fine-tuning in the *Norm* experiment. The values in each cell represent the percentage improvement of the corresponding metric calculated over 100 samples from that distribution.

- *Covariate-Underutilization Experiment* (Can models prioritize exogenous "leading" indicators over historical target noise?): Here, we use 100,000 samples per distribution with a $1\%$ split for faster validation. This experiment tests if fine-tuning can overcome the "multivariate blindness" where models ignore highly predictive exogenous signals in favor of univariate persistence. The results in Table 7 and Fig. 10 show a clear aggregate improvement across all three metrics.

- *All-in-One Experiment* (How does a unified training objective affect performance across heterogeneous Simple-TimeBench data distributions?): Here, we use all the SimpleTSBench dataset to assess the possibility of learning covariate usage without sacrificing other data performance. Similar to the previous experiment, 100,000 samples per distribution with a $1\%$ split is used for faster validation. The results in Table 2 and Fig. 12 show a marginal boost, with negligible improvements in the Leading-Covariate distributions.

For all experimental configurations, models undergo supervised fine-tuning for up to 250,000 optimization steps, $1e-4$ learning rate, and 256 as the batch size. We incorporate an early stopping mechanism that triggers upon a sustained increase in validation loss to mitigate overfitting. Following the training phase, we restore and evaluate the model checkpoint that achieved the minimum validation error to ensure optimal generalization. The comprehensive evolution of these training dynamics across all model architectures is detailed in Fig. 11, which illustrates the complete fine-tuning loss trajectories.

| Family | Distribution | MAPE | | | relMAE$_{mean}$ | | | relMAE$_{last}$ | | |
|---|---|---|---|---|---|---|---|---|---|---|
| | | Base | Fine-tuned | Improvement % | Base | Fine-tuned | Improvement % | Base | Fine-tuned | Improvement % |
| I.I.D. Noise | binary | 0.25 | 0.17 | 32.83 | 0.77 | 0.54 | 30.02 | 0.73 | 0.51 | 30.02 |
| I.I.D. Noise | cauchy | 2.37 | 22.89 | −865.51 | 0.86 | 0.98 | −13.94 | 0.82 | 0.93 | −13.94 |
| I.I.D. Noise | laplace | 1.53 | 6.35 | −314.98 | 1.00 | 0.78 | 21.78 | 0.63 | 0.50 | 21.78 |
| I.I.D. Noise | lognormal | 0.52 | 0.45 | 12.75 | 0.94 | 0.71 | 24.58 | 0.73 | 0.55 | 24.58 |
| I.I.D. Noise | normal | 1.22 | 5.58 | −357.85 | 1.00 | 0.72 | 28.13 | 0.74 | 0.53 | 28.13 |
| I.I.D. Noise | poisson | 0.52 | 0.43 | 17.68 | 0.98 | 0.74 | 24.22 | 0.72 | 0.55 | 24.22 |
| I.I.D. Noise | skew-normal | 9.71 | 8.10 | 16.57 | 0.99 | 0.74 | 25.17 | 0.69 | 0.51 | 25.17 |
| I.I.D. Noise | student-t | 1.29 | 3.86 | −198.50 | 1.00 | 0.75 | 24.69 | 0.71 | 0.54 | 24.69 |
| I.I.D. Noise | uniform | 1.19 | 2.93 | −146.16 | 1.00 | 0.72 | 28.04 | 0.76 | 0.55 | 28.04 |
| Drift/Shocks | gbm | 0.58 | 0.41 | 29.14 | 0.92 | 0.73 | 20.69 | 0.99 | 0.79 | 20.69 |
| Drift/Shocks | impulse | 1.00 | 0.31 | 68.49 | 0.52 | 0.53 | −2.81 | 0.44 | 0.46 | −2.81 |
| Drift/Shocks | intermittent | 1.00 | 0.42 | 57.92 | 0.58 | 0.57 | 0.92 | 0.51 | 0.51 | 0.92 |
| Drift/Shocks | logistic | 0.88 | 0.48 | 45.60 | 0.99 | 0.63 | 35.77 | 0.77 | 0.49 | 35.77 |
| Drift/Shocks | piecewise-constant | 1.92 | 1.05 | 45.29 | 0.48 | 0.44 | 8.74 | 1.19 | 1.09 | 8.74 |
| Drift/Shocks | random-walk | 3.12 | 1.04 | 66.75 | 0.41 | 0.31 | 24.74 | 1.02 | 0.77 | 24.74 |
| Median | | 1.19 | 1.04 | **17.68** | 0.94 | 0.72 | **24.58** | 0.73 | 0.54 | **24.58** |

*Table 7.* Performance gains from fine-tuning Chronos-2 on in the *Covariate-Underutilization* experiment.

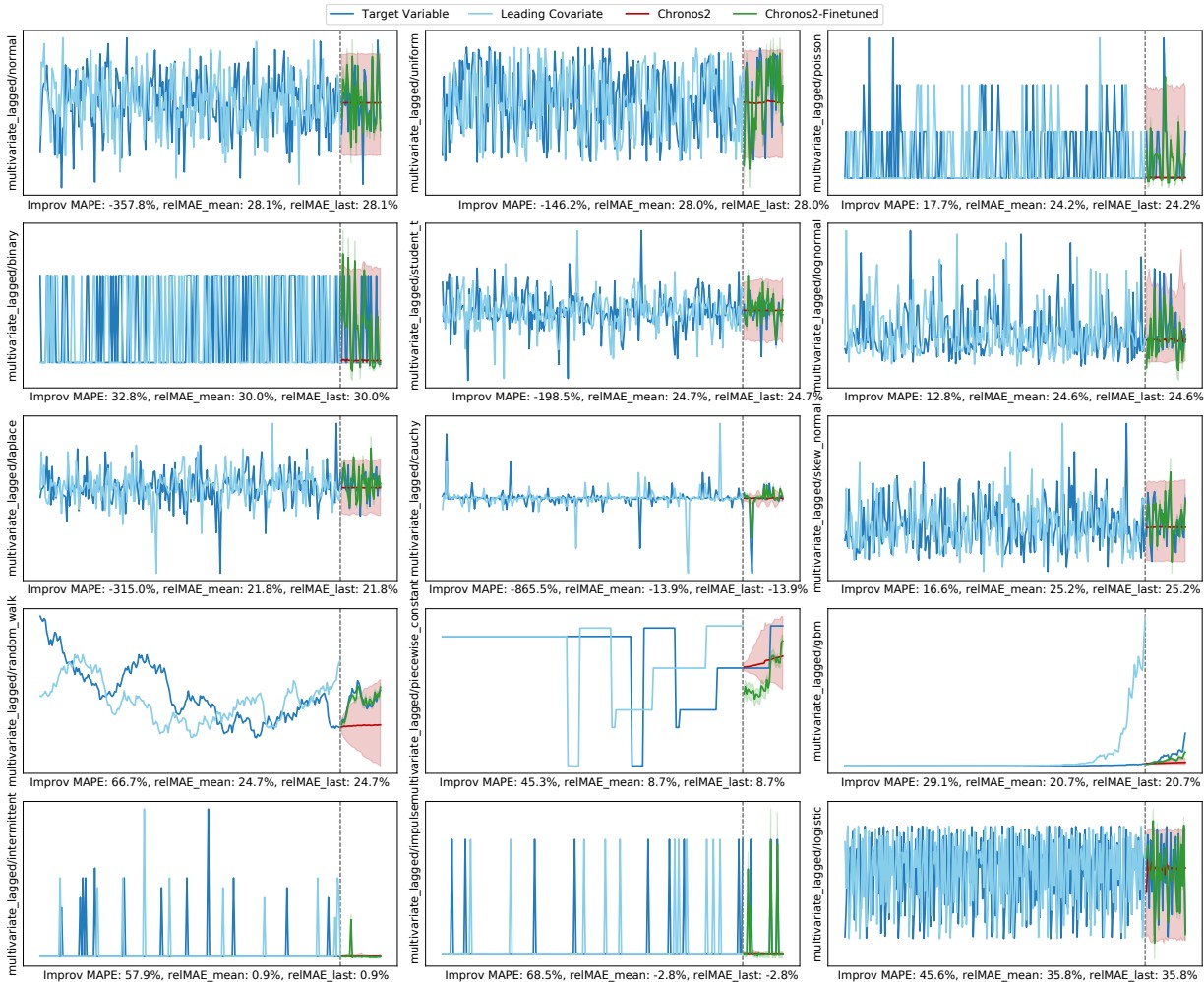

*Figure 10.* Performance of Chronos-2 fine-tuned model on the leading covariate data in the *Covariate-Underutilization* experiment. The values in each cell represent the percentage improvement of the corresponding metric calculated over 100 samples from that distribution.

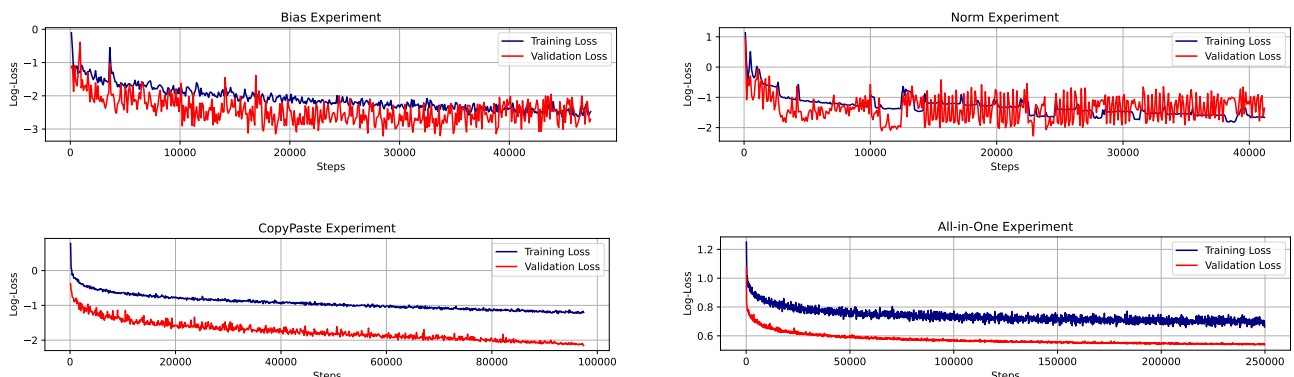

*Figure 11.* Loss trajectories of the four fine-tuning experiments.

It is important to note that the validation loss for the *Covariate-Underutilization* experiment could further go down with continued pretraining and the improvements in utilizing (in this case, copying) informative leading covariate, could further improve. However, for the *All-in-One* experiment undergoes a much longer fine-tuning and reached almost a plateau which we believe is likely an ample window to evaluate the models' learning capacity on SimpleTimeBench. While extended training with hyperparameter tuning or other optimizations might yield marginal performance gains, the structural simplicity of the dataset suggests that an effective architecture should converge on these fundamental patterns within this time frame in the absence of architectural limitations.

## C. GIFT-Eval Univariate vs. Multivariate Experiments

**Experiment Configuration.** Experiments are run leveraging the TimeCopilot (Garza & Rosillo, 2025) setup. The experiments are conducted for all GIFT-Eval datasets across short, medium, and long forecast horizons. The context length for each model follows the specifications provided in the official GIFT-Eval repository's notebooks for the respective models.

**Result Post-processing.** Results only include datasets that have more than one variate. Metric results that are significantly different (lower by 90% or higher than 200%) from the official GIFT-Eval results for a given model on a certain dataset are treated as outliers and excluded from the final reporting to ensure fair comparison.

**Per-Dataset Performance.** Figure 13 shows the univariate versus multivariate performance breakdown by model and dataset, illustrating the inconsistent benefits of multivariate mode across different datasets.

## D. Real-world Datasets Experiments

**Dataset description:**

*Delaware River dataset*: This dataset consists of hourly streamflow discharge measurements for the period 2024-03-05 to 2024-09-09 obtained from two USGS gauges: Montague, NJ (01438500) as the upstream covariate and Trenton, NJ (01463500) as the downstream target. Data were accessed via the dataretrieval Python package (Hodson et al., 2023), which interfaces with the USGS National Water Information System (NWIS) (U.S. Geological Survey, 2024). We performed a lag-correlation analysis which identified a peak cross-correlation at 14 hours. This propagation delay represents a deterministic physical leading indicator where upstream surges directly indicates the downstream flow. Consequently, we set the forecast horizon to 14, ensuring the model must utilize the covariate to accurately predict the target over the full span of the physical propagation. The raw 15-minute data were resampled to hourly means, and gaps (e.g. maintenance intervals) were forward-filled to maintain a continuous hourly timeline for model compatibility.

*NVIDIA dataset*: This dataset comprises daily historical stock data for NVIDIA (NVDA) covering the period from 2020-01-02 to 2025-12-31. The close amount is used as the primary target variable. To isolate the models' ability to attend to explicit covariate signals, we constructed a synthetic leading covariate by shifting the target price series forward by 16 days. Because financial markets are closed on weekends and public holidays, the raw trading data contains irregular temporal

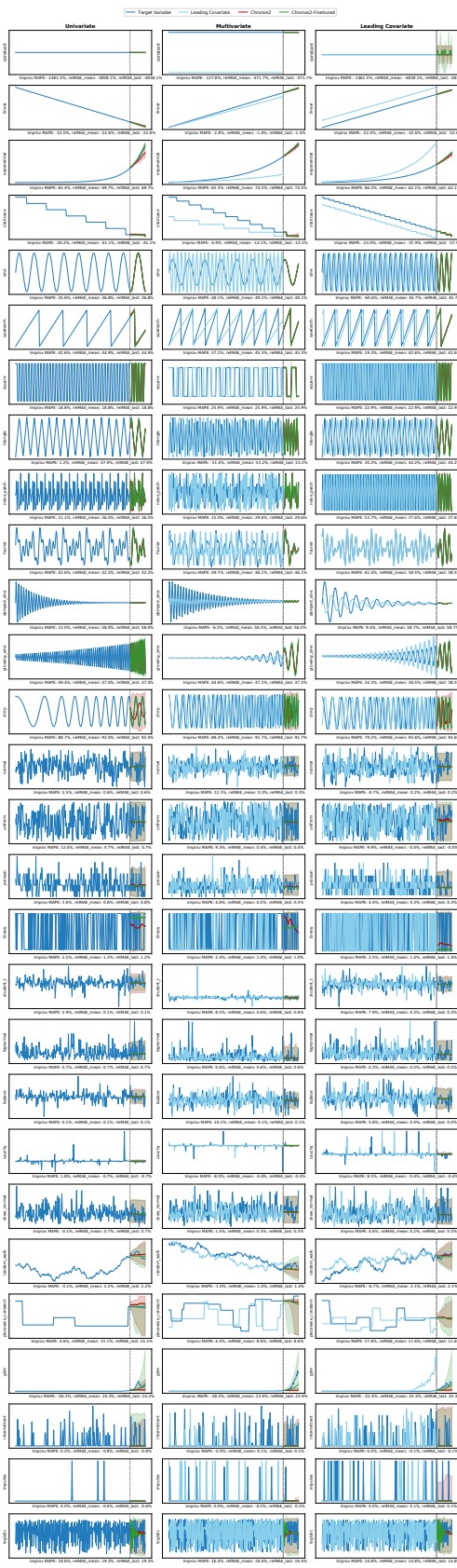

*Figure 12.* Performance of Chronos-2 fine-tuned model on the leading covariate data in the *All-in-One* experiment. The values in each cell represent the percentage improvement of the corresponding metric calculated over 100 samples from that distribution.

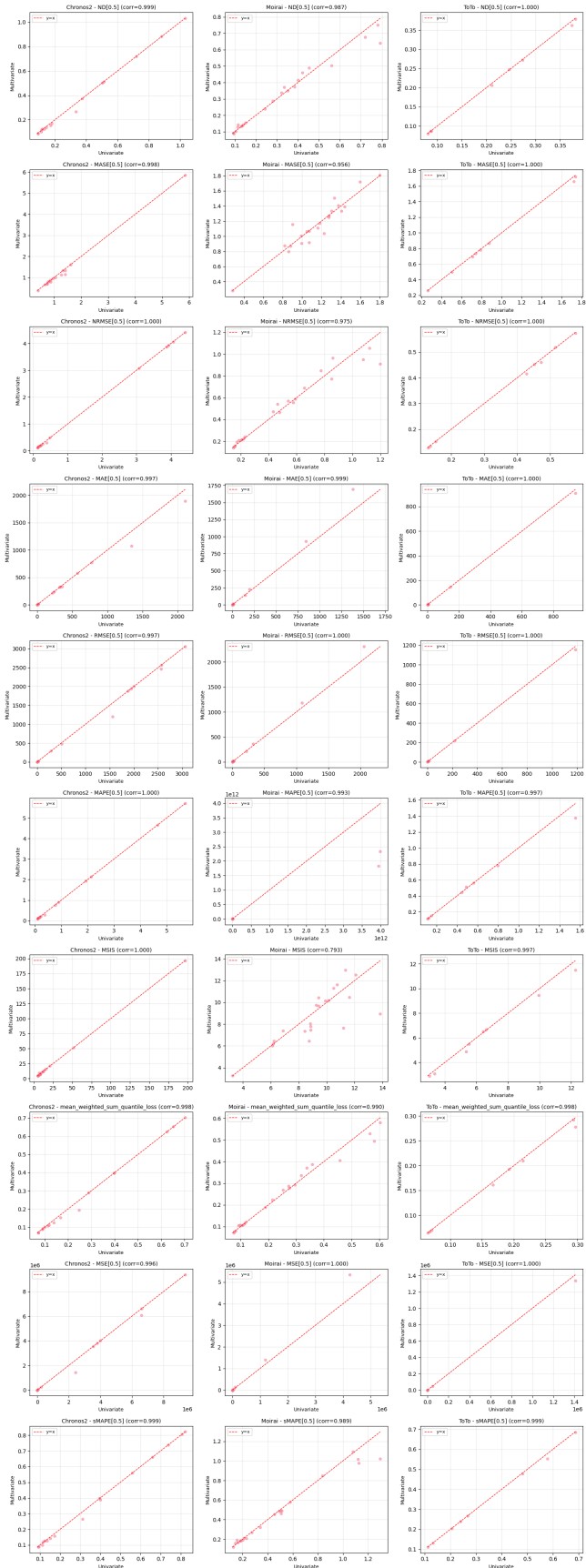

*Figure 13.* Univariate vs. Multivariate performance by model and metric on GIFT-Eval. Each point represents a single dataset/term experiment.

gaps that can impede frequency inference in foundation models like Chronos-2. To ensure a continuous daily frequency, we performed a forward-fill operation for all missing calendar days, effectively maintaining the last known closing price until the next trading session. The forecast horizon was set to 16, matching the leading indicator's displacement and requiring the model to retrieve the future value directly from the covariate channel.

**Experiment setup:**

To evaluate the models, we employed an expanding-window evaluation strategy, specifically designating the final $20\%$ of each dataset as the held-out test set. Within this test block, we used a fold length equal to the forecast horizon ($H$) to ensure independent, non-overlapping segments. For the Delaware River dataset, we set $H = 14$ to align with the physical peak lag, while for NVIDIA, we set $H = 16$ to match the leading indicator. We compared Chronos-2 in both univariate and covariate-support modes against a DLinear baseline. To ensure a competitive DLinear baseline, we performed a grid search hyper parameter tuning over lookback windows of $\{0.5 * H, 1H, 2H, 4H\}$ using the $10\%$ of data immediately preceding the test set as a validation set, selecting the window with the lowest average validation MSE.

**Detailed Results:**

Table 8 presents the performance metrics for both models across the two real world datasets, detailing the shift from univariate forecasting to covariate-supported forecasting. To provide a comprehensive view of model behavior, we include performance improvements across three key metrics: Mean Squared Error (MSE), Mean Absolute Percentage Error (MAPE), and Mean Absolute Scaled Error (MASE). Note that for the MASE calculation, we used a seasonality of 1, treating the series as non-seasonal to focus on the immediate step-by-step predictive gains.

A clear and sharp difference in performance is visible: in the Univariate mode, Chronos-2 consistently outperforms the DLinear baseline. This is especially evident on the Delaware River dataset, showcasing its strong pre-trained understanding of general time series patterns. However, this situation completely reverses once covariate support is introduced. While Chronos-2 shows only marginal improvements (ranging from $5.9\%$ to $6.3\%$ in MASE), DLinear demonstrates a superior ability to leverage the informative signals. This results in substantial MASE reductions of $24.1\%$ on the river flow and a dramatic $62.0\%$ on the NVIDIA leak. Consequently, despite being a simpler model, DLinear with covariate support outperforms Chronos-2 in the final results, highlighting the foundation model's struggle to prioritize specific leading indicators over its historical target priors.

To further investigate the localized impact of leading covariates, we include a zoomed-in visualization of the primary peaks in the test period of both datasets, as shown in Figure 14. For the Delaware River dataset, the view covers $\pm 100$ hours around the highest peak, while the NVIDIA dataset displays $\pm 30$ days. These granular views confirm that while Chronos-2 exhibits superior baseline performance in the univariate setting by closely following the target's general magnitude, it remains largely indifferent to the covariate signal during rapid transitions. Even with covariate support, Chronos-2's forecast remains muted and delayed, essentially mirroring its own univariate prediction. In contrast, DLinear undergoes a dramatic shift when provided with covariate support: its prediction curve aligns almost perfectly with the leading edge of the covariate spikes. This is particularly evident in the NVIDIA leak test, where DLinear effectively transforms from a lagging, noisy estimator into a precise, time-aligned forecaster. These visual results reinforce the conclusion that DLinear successfully internalizes the deterministic relationship between the covariate and the target, whereas Chronos-2 relies on smoothed historical target patterns, resulting in significant phase lag during critical spikes.

*Table 8.* Univariate and with covariate support detailed forecast metrics comparison of Chronos-2 v.s. DLinear on Delaware River and NVIDIA datasets

| Dataset | Model | Metric | Univariate | With Covariate | Improvement |
|---------|-------|--------|-----------|----------------|-------------|
| Delaware River | Chronos-2 | MSE | 8.99e+06 | 8.05e+06 | +10.5% |
| | | MAPE | 6.05 | 5.88 | +2.8% |
| | | MASE | 7.16 | 6.74 | +5.9% |
| | DLinear | MSE | 1.28e+07 | 6.05e+06 | +52.7% |
| | | MAPE | 7.16 | 6.07 | +15.2% |
| | | MASE | 8.82 | 6.69 | +24.1% |
| NVIDIA | Chronos-2 | MSE | 65.76 | 57.79 | +12.1% |
| | | MAPE | 4.32 | 4.05 | +6.3% |
| | | MASE | 10.56 | 9.90 | +6.3% |
| | DLinear | MSE | 71.76 | 11.47 | +84.0% |
| | | MAPE | 4.67 | 1.77 | +62.1% |
| | | MASE | 11.29 | 4.29 | +62.0% |

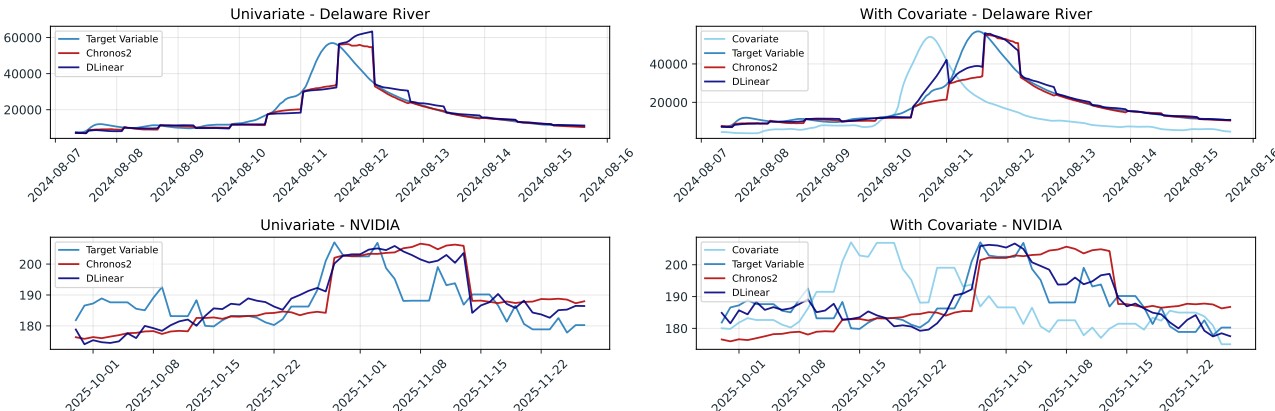

*Figure 14.* Univariate and with covariate support forecast comparison of Chronos-2 v.s. DLinear on Delaware River and NVIDIA datasets zooming into critical spikes in the test period

