# OpenReview forum: "Foundations without Fundamentals: Zero-Shot Blind Spots in Time Series FMs"
_ICML.cc/2026/Workshop/FMSD — FMSD @ ICML 2026 SpotlightOral_

### Official Review · Reviewer_bjyG · 2026-05-19
**Strong diagnostic study of TSFM blind spots**

**Rating:** 9
**Confidence:** 4

**Review:**

## Summary

This paper studies blind spots in the evaluation of time-series foundation models (TSFMs). The main argument is that although current TSFMs achieve strong performance on broad aggregate benchmarks, such benchmarks may fail to reveal whether models have internalized basic temporal logic. Inspired by diagnostic evaluation approaches used for large language models, the authors introduce **SimpleTimeBench**, a unit-test-style benchmark designed to probe simple but fundamental forecasting capabilities.

Using SimpleTimeBench, the paper identifies three major failure modes in current TSFMs: **Mean-Reversion Bias**, **Anti-Exponential Bias**, and **Covariate Underutilization**. Among these, the paper further investigates covariate underutilization through fine-tuning experiments and real-world covariate-supported forecasting tasks. Overall, the paper provides a timely and useful diagnostic perspective on TSFMs. It argues that strong aggregate benchmark performance does not necessarily imply robust foundational temporal capabilities, and it offers meaningful insights into how current TSFMs may fail on simple, interpretable, and practically important forecasting scenarios.

## Strengths

1. **Timely and important problem formulation**
   The paper addresses a highly relevant issue for the TSFM community: whether strong performance on large-scale aggregate benchmarks is sufficient evidence of foundational forecasting capability. The framing is clear and compelling, and the analogy to blind-spot evaluation in LLMs is appropriate and useful.

2. **Well-motivated diagnostic benchmark**
   SimpleTimeBench is a valuable contribution. Rather than proposing another leaderboard-style benchmark, the paper focuses on unit-test-like diagnostics for basic temporal primitives such as trends, periodic signals, stochastic processes, and leading covariates. This makes the benchmark interpretable and useful for understanding specific model failure modes.

## Areas for Improvement

1. **The DLinear comparison is compelling but under-specified**
   The real-world comparison with DLinear is one of the paper's strongest pieces of evidence for covariate underutilization. However, the paper should clarify exactly how DLinear incorporates covariates. DLinear is better described as a channel-independent multivariate forecasting model rather than a purely univariate model, and in its standard form it is not obvious whether a covariate channel can directly influence the target forecast. This ambiguity affects the interpretation of the real-world covariate experiments.

## Detailed Comments

1. **Extend the covariate usage validation beyond Chronos-2 if possible**
   The validation showing that Chronos-2 can use future covariates is useful. However, the main benchmark compares Chronos-2, Moirai, and Toto. Similar validation checks for Moirai and Toto would strengthen the conclusion that the observed failures reflect covariate underutilization rather than model-specific interface mismatch or implementation details.

2. **Clarify the DLinear covariate implementation**
   The DLinear baseline plays an important role in the real-world experiments, especially in showing that a simple model can exploit leading covariates more effectively than Chronos-2. However, the paper should specify exactly how DLinear is configured for covariate-supported forecasting. My understanding is that standard DLinear is often used in a channel-independent manner. The `individual` configuration typically controls whether channels have separate temporal linear layers, but it does not necessarily imply cross-channel mixing from covariates to the target forecast. If the authors use a modified or covariate-augmented DLinear implementation, they should explicitly describe how the target and covariate channels are represented and whether cross-channel mappings are learned. Without this clarification, the reported DLinear gains with covariate support are difficult to fully interpret and reproduce.

3. **Discuss related work on compositional reasoning in TSFMs**
   The paper would benefit from discussing recent work on compositional reasoning in time-series foundation models, such as *Investigating Compositional Reasoning in Time Series Foundation Models*(https://arxiv.org/abs/2502.06037). This line of work is closely related in spirit, as it also questions whether strong zero-shot benchmark performance implies genuine temporal reasoning or generalization ability.

## Justification of Score

I view this as a strong workshop submission. The paper identifies an important gap in how TSFMs are currently evaluated and proposes a clear, interpretable, and useful diagnostic benchmark. The experimental narrative is well structured, and the identified failure modes are meaningful for both researchers and practitioners. In particular, the analysis of covariate underutilization is a valuable contribution because many real-world forecasting problems rely on informative exogenous variables.

The main issue is that the DLinear comparison, which is central to the real-world covariate experiments, needs a more precise methodological description. Nevertheless, the paper makes a timely and useful contribution to the workshop. It provides a compelling diagnostic perspective on TSFMs and raises important questions about whether current models truly possess robust foundational temporal capabilities. My overall recommendation is positive.

---

### Official Review · Reviewer_8aN1 · 2026-05-20
**Foundations without Fundamentals: Zero-Shot Blind Spots in Time Series FMs**

**Rating:** 6
**Confidence:** 4

**Review:**

The authors introduce SimpleTimeBench, a diagnostic suite testing Time Series Foundation Models (TSFMs) on fundamental temporal logic, including deterministic trends, periodic signals, and leading indicators. Evaluations of Chronos-2, Moirai, and Toto show significant zero shot blind spots, such as mean-reversion bias, anti-exponential bias, and an inability to exploit perfect leading covariates. Fine tuning experiments and real world case studies (Delaware River and NVIDIA stock) were done and they confirm that current TSFMs struggle with these basic primitives, often underperforming simple baselines like DLinear when highly informative covariates are available.

Strengths:
1. Moving from aggregate performance metrics on complex data to "unit testing" basic temporal reasoning is a good contribution.
2. Including real world datasets effectively proves that these synthetic blind spots translate into practical forecasting failures also.

Areas for Improvement:
1.  The fine tuning experiments testing capacity bottlenecks do it for Chronos-2. Should extend to Toto or Moirai.
2. ToTo achieves a 100% win rate against its univariate baseline even though there were only limited absolute metric improvements. Should talk more about this behaviour.

---

### Official Review · Reviewer_HTQH · 2026-05-22
**Strong diagnostic benchmark for time-series FMs, with a few claims that should be softened**

**Rating:** 7
**Confidence:** 4

**Review:**

### Summary

This paper introduces SimpleTimeBench, a diagnostic benchmark for testing basic temporal reasoning in time-series foundation models. The suite covers univariate, multivariate, and past leading-covariate settings, with simple processes such as linear and exponential trends, periodic signals, IID noise, random walks, and shifted covariates. The authors evaluate Chronos-2, Moirai, and Toto, and show several zero-shot blind spots: forecasts often revert toward the historical mean, exponential growth is not extrapolated well, and highly informative leading covariates are not reliably used. They also fine-tune Chronos-2 on targeted subsets, compare univariate and multivariate behavior on GIFT-Eval, and include covariate-focused studies on Delaware River streamflow and an NVIDIA shifted-target stress test.

### Strengths

- The paper is well matched to the workshop because it studies whether structured-data foundation models actually learn basic time-series behavior.
- I like the “unit test” framing. It makes the benchmark more interpretable than another aggregate leaderboard and gives a clearer way to debug model failures.
- The failure cases are organized clearly. The paper separates mean-reverting behavior, weak exponential extrapolation, and poor use of leading covariates.
- The diagnostic metrics, including relative MAE, MAR, PVR, and MBR, are useful because they expose behavior that a single forecasting score would hide.
- The Delaware River example is a helpful real-world case where the covariate relation has a physical interpretation. The NVIDIA example is also useful, although mainly as a controlled stress test.

### Weaknesses

- Some claims about architectural limitations are stronger than the evidence fully supports. The all-in-one fine-tuning result suggests a real limitation, but it does not completely separate architecture, optimization, data mixture, and training procedure.
- The benchmark would be stronger with more simple baselines across the synthetic tasks, such as linear regression, ARIMA/ETS where appropriate, DLinear/NLinear, and lag-copy or oracle-style baselines for covariate tasks.
- The NVIDIA setup uses a shifted version of the target as the covariate, so it should be described more carefully as a controlled diagnostic rather than a natural real-world leading indicator.
- I would like to see more reporting over random seeds, generated datasets, or context-length choices, since the benchmark depends on synthetic generation.

### Suggestions

The authors should soften the architectural-bottleneck claim and frame it as evidence that current training and modeling setups do not reliably learn these primitives together. Adding simple and oracle-style baselines would make the severity of the covariate failures clearer. I also recommend being more explicit about the distinction between unrelated past covariates, shifted past leading covariates, and future-known covariates.

### Justification of Score

I recommend clear accept. The paper is timely, readable, and strongly relevant to the workshop. Even if some broader claims should be toned down, the benchmark and empirical findings are useful for researchers building or evaluating time-series foundation models.